# Egyptian Fruit Bats (*Rousettus aegyptiacus*) Were Resistant to Experimental Inoculation with Avian-Origin Influenza A Virus of Subtype H9N2, But Are Susceptible to Experimental Infection with Bat-Borne H9N2 Virus

**DOI:** 10.3390/v13040672

**Published:** 2021-04-14

**Authors:** Nico Joel Halwe, Marco Gorka, Bernd Hoffmann, Melanie Rissmann, Angele Breithaupt, Martin Schwemmle, Martin Beer, Ahmed Kandeil, Mohamed A. Ali, Ghazi Kayali, Donata Hoffmann, Anne Balkema-Buschmann

**Affiliations:** 1Institute of Diagnostic Virology, Friedrich-Loeffler-Institut, 17493 Greifswald, Insel Riems, Germany; nico.halwe@fli.de (N.J.H.); marco.gorka@fli.de (M.G.); Bernd.Hoffmann@fli.de (B.H.); Martin.Beer@fli.de (M.B.); 2Institute of Novel and Emerging Infectious Diseases, Friedrich-Loeffler-Institut, Insel Riems, 17493 Greifswald, Insel Riems, Germany; Melanie.Rissmann@fli.de (M.R.); anne.buschmann@fli.de (A.B.-B.); 3Department of Experimental Animal Facilities and Biorisk Management, Friedrich-Loeffler-Institut, 17493 Greifswald, Insel Riems, Germany; angele.breithaupt@fli.de; 4Institute of Virology, Medical Center-University of Freiburg, 79104 Freiburg, Germany; martin.schwemmle@uniklinik-freiburg.de; 5Faculty of Medicine, University of Freiburg, 79104 Freiburg, Germany; 6Center of Scientific Excellence for Influenza Viruses, National Research Centre, Dokki, Giza 12311, Egypt; Ahmed.Kandeil@human-link.org (A.K.); Mohamed.Ali@human-link.org (M.A.A.); 7Department of Epidemiology, Human Genetics and Environmental Sciences, University of Texas Health Sciences Center, Houston, TX 77030, USA; ghazi@human-link.org; 8Human Link, Dubai, United Arab Emirates

**Keywords:** influenza virus, H9N2, *Rousettus aegyptiacus*, avian-related H9N2, bat-related H9N2

## Abstract

Influenza A viruses (IAV) of subtype H9N2, endemic in world-wide poultry holdings, are reported to cause spill-over infections to pigs and humans and have also contributed substantially to recent reassortment-derived pre-pandemic zoonotic viruses of concern, such as the Asian H7N9 viruses. Recently, a H9N2 bat influenza A virus was found in Egyptian fruit bats (*Rousettus aegyptiacus*), raising the question of whether this bat species is a suitable host for IAV. Here, we studied the susceptibility, pathogenesis and transmission of avian and bat-related H9N2 viruses in this new host. In a first experiment, we oronasally inoculated six Egyptian fruit bats with an avian-related H9N2 virus (A/layer chicken/Bangladesh/VP02-plaque/2016 (H9N2)). In a second experiment, six Egyptian fruit bats were inoculated with the newly discovered bat-related H9N2 virus (A/bat/Egypt/381OP/2017 (H9N2)). While *R. aegyptiacus* turned out to be refractory to an infection with H9N2 avian-type, inoculation with the bat H9N2 subtype established a productive infection in all inoculated animals with a detectable seroconversion at day 21 post-infection. In conclusion, Egyptian fruit bats are most likely not susceptible to the avian H9N2 subtype, but can be infected with fruit bat-derived H9N2. H9-specific sero-reactivities in fruit bats in the field are therefore more likely the result of contact with a bat-adapted H9N2 strain.

## 1. Introduction

Low-pathogenic avian influenza viruses (LPAIV) of subtype H9N2 are globally widespread in poultry and endemic in China and Eurasian countries [1,2]. While causing severe losses in poultry production, spill-over infections of H9N2 and its reassortants pose a constant threat to the human population [3,4,5]. In 1999, the first isolation of H9N2 from two diseased children in Hong Kong was published [6]. In addition, H9N2 viruses are able to replicate in further mammalian species, such as pigs or mice [7]. Multiple cases of H9N2 outbreaks in avian species in Africa have been reported inter alia from Egypt [8,9], where the virus is endemic [2].

The first evidence of influenza A and influenza A-like viruses originating from different bat species was found in 2012 and 2013: two novel influenza A-like viruses, provisionally designated as H17N10 and H18N11, were identified via next-generation sequencing in the feces of a little yellow-shouldered fruit bat (*Sturnira lilium*) and a South American flat-faced fruit-eating bat (*Artibeus planirostris*), respectively [10,11]. These findings raised the question of bats in general being a reservoir host for multiple influenza A viruses. Most importantly, however, these findings opened questions about an evolutionary relationship between avian and bat-borne influenza A viruses. Furthermore, H9-specific sero-reactivity was demonstrated in 30% of straw-colored fruit bats (*Eidolon helvum*) in Ghana [12]. Furthermore, only recently, a new distinct influenza A virus was found and isolated from Egyptian fruit bats (*Rousettus aegyptiacus*) in Egypt. Interestingly, the hemagglutinin (HA) and neuraminidase (NA) encoding segments of this particular virus were closely related to avian H9N2-viruses [13,14].

Considering the findings of influenza A and influenza A-like viruses and specific antibodies in multiple bat species, bats in general might be involved in influenza A virus transmission cycles and might represent a potential vessel for reassortment processes [14,15]. However, to what extent Egyptian fruit bats support infection with IAV, including the newly described bat H9N2 subtype, remains to be shown.

In order to specifically investigate susceptibility, pathogenesis and virus shedding, we oronasally infected two groups of Egyptian fruit bats (*Rousettus aegyptiacus*) with either a poultry-origin H9N2 strain of clade G1 (A/layer chicken/Bangladesh/VP02-plaque/2016) or the fruit bat-derived strain A/bat/Egypt/381OP/2017 (H9N2). The latter group was cohoused with naïve contact bats (24 h post-infection) to further elucidate H9N2 bat flu virus transmission.

## 2. Materials and Methods

### 2.1. Viruses

LPAIV H9N2 of clade G1, namely A/layer chicken/Bangladesh/VP02-plaque/2016 (accession numbers EPI1807182-1807189), was obtained from the Friedrich-Loeffler-Institut (FLI) virus repository [9]. The virus stock was propagated in Madin-Darby Canine Kidney (MDCK) type II cells (Collection of Cell Lines in Veterinary Medicine CCLV RIE1061) for two days at 37 °C under 5% CO2 atmosphere using a mixture of equal volumes of Eagle Minimum Essential Medium (MEM) (Hank’s balanced salts solution) and Eagle MEM (Earle’s balanced salts solution), 2 mM L-Gln, nonessential amino acids, adjusted to 850 mg/L NaHCO3, 120 mg/L sodium pyruvate, pH 7.2 in the presence of tosylsulfonyl phenylalanyl chloromethyl ketone (TPCK)-treated trypsin (Sigma, Munich, Germany). The LPAIV H9N2 A/bat/Egypt/381OP/2017 virus isolate (accession numbers MH376902.1-MH376909.1) was propagated via infection of embryonated chicken eggs for 72 h. Hereafter, the allantoic fluid was harvested and stored at −80 °C. This virus stock was once more propagated in cell culture on CCLV RIE1495 MDCK cells for two days at 37 °C, 5% CO2 using the MEM mixture mentioned above. Viral supernatants of both avian and bat H9N2 were harvested, cleared of debris and stored at −80 °C. Viral titers were determined by virus titration and calculated by the Reed–Muench method [16].

### 2.2. Experimental Inoculation and Sampling of Bats

Overall, all 20 Egyptian fruit bats (*Rousettus aegyptiacus*) from both experiments were reared at FLI and tested negative for influenza A reactive antibodies via immunofluorescence analysis (IFA) using the inoculation viruses. Six bats in each experiment were anesthetized by inhalation of 5% isoflurane in O2 and oronasally inoculated with 100 µL H9N2 (A/layer chicken/Bangladesh/VP02-plaque/2016) distributed equally into each nostril and mouth at a concentration of 10^4.7^ TCID_50_ per animal (Figure 1) or with 100 µL 10^4.7^ TCID_50_ per animal of A/bat/Egypt/381OP/2017 (H9N2) (Figure 2), respectively. The remaining two animals in the first experiment served as negative controls and were mock-infected with the same volume of PBS (Figure 1). The six naïve animals in the second experiment (Figure 2) were co-housed with the inoculated animals 24 h post-infection. Clinical signs (nasal discharge, reduced activity, neurological symptoms and dyspnea) were monitored daily. In the second experiment, all infected and two contact animals carried a temperature transponder (Anipill, Bodycap, Hérouville Saint-Clair, France) that had been surgically implanted under general anesthetics into the abdominal cavity two weeks prior to infection. The transponder automatically transmitted the body core temperature to the monitor device every 15 min. In addition, rectal temperature, body weight and oral and rectal swabs were collected at two-day intervals using plain swab sterile paper applicator cotton tipped 164C, Copan, Brescia, Italy. The swabs were transferred to 2 mL of cell culture medium containing 1% Baytril (Bayer, Leverkusen, Germany), 0.5% Lincomycin (WDT, Garbsen, Germany) and 0.2% Amphotericin/Gentamycin (Fisher Scientific Waltham, MA, USA), followed by 30 min. incubation at room temperature under roughly shaking at 300 rpm (KS 260 control, IKA, Staufen, Germany). In the first experiment, all bats were sacrificed 21 dpi, whereas in the second experiment, two inoculated and two naïve animals were sacrificed at day seven, while the residual eight animals were kept until day 21. At necropsy, samples were obtained from the following organs: conchae, trachea, lung (left caudal lung lobe and right cranial lung lobe), heart, kidney, liver, spleen, duodenum, jejunum, colon, rectum, skeletal muscle and thigh muscles. One part of each tissue was fixed in 10% neutral-buffered formalin and another part was stored at −80 °C until further processing. Blood samples for serology were collected from final bleed during the euthanasia procedure.

### 2.3. Organ Homogenization

Organ pieces (approximately 2 × 2 × 2 mm) were transferred into 2 mL collection tubes prepared with a stainless-steel bead (diameter 5 mm) and 1 mL of DMEM supplemented with antibiotics (1% penicillin-streptomycin, Biochrome, Berlin, Germany). Homogenization was performed using a TissueLyser II instrument (Qiagen, Hilden, Germany) for 2 min at 300 Hz. Supernatants for RNA extractions were acquired following centrifugation at 13,000 rpm for 2 min.

### 2.4. RNA Isolation

RNA extraction of all oral and rectal swabs, as well as organ samples after euthanasia from both experiments was performed via the NucleoMag Vet kit (Macherey-Nagel, Düren, Germany) according to manufacturer’s instructions on a Biosprint 96 platform (Qiagen, Hilden, Germany).

### 2.5. Real-Time RT-PCR (RT-qPCR)

RT-qPCRs of organs, oral and rectal swabs in the first experiment (avian origin H9N2) were performed as described before [17].

In order to detect A/bat/Egypt/381OP/2017 (H9N2) viral RNA in the second experiment, a specific primer and probe system was designed (Table 1). For process control, a genomic nucleic acid was co-amplified in the PCR runs using the HEX channel [18]. The final composition of the RT-qPCR reactions was 1.75 μL of RNase-free water, 6.25 μL of 2x qScript XLT One-Step RT-qPCR ToughMix (Quanta, Beverly, MA, USA), 1 μL of primer-probe-mix-FAM, 1 μL of beta-actin DNA-mix2-HEX and 2.5 µL of template RNA. All RT-qPCRs were run on the CFX 96 real-time PCR cycler (Bio-Rad, Hercules, CA, USA). The temperature profile used was 10 min at 50 °C (reverse transcription), 1 min at 95 °C (inactivation of the reverse transcriptase/activation Taq polymerase) followed by 45 cycles of 10 s at 95 °C (denaturation), 30 s at 57 °C (annealing) and 30 s at 68 °C (elongation). Fluorescence values (FAM, HEX) were collected during the annealing step. Absolute quantification was done using a standard of known concentrations, corresponding to the RNA of the original virus used for inoculation. Quantification was established by the QX200 Droplet Digital PCR System in combination with the 1-Step RT-ddPCR Advanced Kit for Probes (BioRad, Hercules, CA, USA).

### 2.6. Histopathology

Based on RT-qPCR data, selected organs from bats sacrificed at 7 dpi were trimmed for paraffin-embedding: nasal atrium (non-respiratory region), nasal conchae (respiratory and olfactory region), duodenum, jejunum, colon and rectum of inoculated #5 and #6, as well as transmission #11 and #12. Tissue sections were stained with hematoxylin and eosin (HE). Immunohistochemistry for viral antigen detection using a primary antibody against the M protein of IAV (ATCC clone HB-64) was performed on consecutive slides as described earlier [19]. The severity of lesions was recorded on an ordinal scoring scale with scores 0 = no lesion, 1 = minimal or <5% per affected cells/tissue slide; 2 = mild or 6–40% affected; 3 = moderate or 41–80% affected; and 4 = severe or > 80% affected. The distribution of antigen was semi-quantitatively scored on an ordinal 0–4 scale: 0 = negative; 1 = focal or oligofocal, 2 = multifocal, 3 = coalescing and 4 = diffuse immunoreactive cells. Evaluation and interpretation were performed by a board-certified pathologist (DiplECVP).

### 2.7. Propagation of A/Bat/Egypt/381OP/2017 (H9N2) Virus Isolates from Bat Samples

Animal samples were initially diluted in medium (see section “Experimental inoculation and sampling of bats”), whereupon 200 µL was transferred into the allantoic cavity of embryonated chicken eggs (three eggs per sample) followed by incubation for five days at 37 °C. Afterwards, the eggs were placed at −80 °C for 30 min to ensure the death of the chicken embryo, whereupon the egg was opened to allow harvesting of the allantoic fluid. The fluid was centrifuged at 3500 rpm for 10 min, followed by a TRIzol LS reagent (life technologies, Carlsbad, CA, USA)-based RNA extraction (according to manufacturer’s instructions). Existence of A/bat/Egypt/381OP/2017 (H9N2) viral RNA and thereby evidence for successful propagation of A/bat/Egypt/381OP/2017 (H9N2) was finally verified via RT-qPCR as described before.

### 2.8. Serology

All serum samples were heat-inactivated at 56 °C for 30 min and animal sera from both experiments were analyzed by a commercial enzyme-linked immunosorbent assay (ELISA) for the presence of antibodies against the IAV nucleoprotein (NP) according to manufacturer’s instructions (ID-Vet, Montepellier, France). In addition, a virus neutralization assay was performed in the first experiment. In brief, 50 µL of medium containing H9N2 (A/layer chicken/Bangladesh/VP02-plaque /2016) at a concentration of 10^3.3^ TCID_50_/_mL_ was mixed with the same volume of diluted sera. Each serum was prepared in triplicate in a 96-well plate. After incubation for 2 h at 37 °C, the samples were transferred into a second 96-well plate prepared 24 h earlier containing 100 µL medium on 24 h grown MDCK II cells. Viral replication was assessed after an incubation period of 5 days (37 °C, 5% CO2) via visualization of the cytopathic effect. Validation was achieved by titration of the virus dilutions. In the second experiment, seroconversion was investigated via IFA as well. For that reason, MDCK cells were seeded in a 96-well plate and were grown for 24 h. Afterwards, cells were infected with A/bat/Egypt/381OP/2017 (H9N2) for 24 h followed by cell fixation with 4% paraformaldehyde (PFA), as well as permeabilization with 0.5% Triton-X-100 in PBS. Cells were then stained for 1 h with a dilution series from 1:10–1:256 of the appropriate animal serum, followed by three PBS-washing steps and the incubation with a secondary goat-α-bat IgG (H + L) antibody (Novus biologicals, Wiesbaden Nordenstadt, Germany) for 1 h. Subsequently, cells were washed three times with PBS again, followed by the addition of an Alexa Fluor 488-coupled chicken-α-goat IgG (H + L) third antibody for 1 h to allow visualization under the fluorescence microscope after a final three-times PBS-wash procedure.

## 3. Results

### 3.1. Egyptian Fruit Bats Are Resistant to Infection with an Avian H9N2 Strain

Following oronasal inoculation of Egyptian fruit bats (*Rousettus aegyptiacus*) with 10^4.7^ TCID50 per animal of A/layer chicken/Bangladesh/VP02-plaque /2016 (H9N2), neither clinical signs nor any significant changes in body weight or body temperature could be observed during the experiment. In addition, no viral RNA was detected in organs, oral or rectal swabs. Samples below 10 viral genome copy numbers per mL (equivalent to Ct 39.2) were considered negative. At necropsy, no macroscopic lesions were evident. Analysis of sera until 21 dpi by IAV-antibody-ELISA and serum neutralization assay revealed no evidence of seroconversion. Taken together, these observations strongly suggest that this H9N2 subtype of avian origin fails to replicate in Egyptian fruit bats.

### 3.2. Egyptian Fruit Bats Are Susceptible to Natural Infection with Bat-Origin H9N2 A/Bat/Egypt/381OP/2017

After infection with the H9N2 subtype of bat origin, none of the inoculated or contact animals showed any obvious clinical signs of an infection, such as anorexia, depression or signs of a respiratory infection. The temperature monitoring data did not reveal any obvious deviation from the physiological oscillation that ranges mainly between 34 and 41 °C (Figure 3), depending on the time of day and the animals’ activity. While only one of six contact animals (contact animal #4) displayed altogether two temperature maxima reaching 41 °C within 21 dpi, all six inoculated animals showed at least one maximum exceeding 40 °C, with two animals showing 11 temperature peaks > 40 °C (inoculated animal #2) or six peaks > 40 °C (inoculated animal #4) within the same time period. Inoculated animal #2 reached 41 °C twice during the experiment (Figure 3). Notably, contact animal #4 and inoculated animal #2 were kept in the same cage.

Inoculation of Egyptian fruit bats with 100 µL (dose 10^4.7^ TCID_50_) per animal of A/bat/Egypt/381OP/2017 (H9N2) established a productive infection in these animals. At 1 dpi, one out of six oral swabs, as well as two rectal swabs from the inoculated animals, were positive for the H9N2 bat flu viral genome using RT-qPCR (Figure 4 and Figure 5). At 3 dpi, oral swabs of five out of six inoculated animals, and one rectal swab (inoculated animal #10), were positive for viral genome, ranging from 30 up to 4.6 × 10^3^ viral genome copy numbers per mL (Figure 4 and Figure 5). Three oral swabs and one rectal swab of the inoculated animals after 5 dpi and two oral swabs from inoculated animals after 7 dpi, as well as one oral swab of an inoculated animal after 9 dpi were still positive for A/bat/Egypt/381OP/2017 (H9N2) genome. From 11 dpi until the end of the experiment at 21 dpi, all oral and rectal swabs were negative for H9N2 bat flu RNA (Figure 4 and Figure 5).

Nasal conchae samples of two inoculated animals sacrificed at 7 dpi tested positive in both inoculated animals (Figure 6). Moreover, viral RNA was also detected from the trachea samples of both inoculated animals. As summarized in Figure 6, nasal conchae from two of four inoculated animals sacrificed at day 21 of the animal experiment were still positive for viral RNA of H9N2 bat flu.

Fecal samples of animal cage one were positive for A/bat/Egypt/381OP/2017 (H9N2) viral RNA at 4 and 7 dpi and samples from cage two were positive at 3, 4, 6 and 8 dpi, with maximum viral loads of 4 × 10^3^ genome copies per mL detected in the sample collected at 8 dpi. Only one fecal sample collected from cage three was positive at 3 dpi, while the residual samples were negative for A/bat/Egypt/381OP/2017 (H9N2) viral RNA.

Histopathology identified an oligo-focal, minimal to mild rhinitis in all animals evaluated at 7 dpi (inoculated animal #5 and #6) and 6 days post-contact (contact animal #5 and #6). Additionally, inoculated animal #6 showed a moderate transmigration of granulocytes and lymphocytes as well as intraluminal cellular debris within the nasolacrimal duct (Figure 7A). Contact animal #6 exhibited loss of cilia and moderate degeneration of the mucosal epithelium of the nasal-associated lymphoid tissue (Figure 7C). No lesions were observed in the intestinal tract. Viral antigen could not be detected in the nasal cavity or in the intestinal tract using immunohistochemistry. In summary, Egyptian fruit bats can be productively infected with A/bat/Egypt/381OP/2017 (H9N2) resulting in fecal secretion with maximum viral loads at 8 dpi, whereas transmission to contact animals was rather inefficient and did not result in a sustained infection.

### 3.3. A/Bat/Egypt/381OP/2017 (H9N2) Was Effectively Isolated and Propagated from the Experimental Egyptian Fruit Bat Samples

Since it has been shown previously that A/bat/Egypt/381OP/2017 (H9N2) efficiently replicates in embryonated chicken eggs [13], we attempted to propagate the virus from two RT-qPCR-positive Egyptian fruit bat oral swabs (inoculated animal #3 and inoculated animal #6), as well as from one RT-qPCR-positive nasal turbinate sample (inoculated animal #6) using three replicates, respectively.

As shown in Table 2, all three eggs inoculated with the oral swab sample of inoculated animal #3 (initial Ct-value 30.11) were confirmed to contain A/bat/Egypt/381OP/2017 (H9N2) viral RNA with Ct-values ranging from 31 to 34, suggesting that viral replication may have occurred to a certain extent. However, two out of three eggs inoculated either with the oral swab (initial CT value 28.90) or the nasal conchae organ sample of inoculated animal #6 (initial Ct-value 31.74) showed striking evidence for successful viral replication, with Ct-values of 14 and 13, respectively, whereas the third egg replicate of both samples revealed Ct-values of 28 and 34, respectively. In order to demonstrate that even the least positive Ct-value detected from the respective samples out of the three egg replicates was indicative for A/bat/Egypt/381OP/2017 replication, we infected MDCK-cells with 100 µL of the aforementioned egg samples and incubated the cells at 37 °C for three days. As expected, a cytopathic effect (CPE) was observed in all of the three infected samples, thereby further confirming the successful re-isolation after inoculation of the embryonated eggs. Together, these observations indicate that Egyptian fruit bats experimentally infected with A/bat/Egypt/381OP/2017 (H9N2) shed infectious virus.

### 3.4. Egyptian Fruit Bats Seroconvert upon Inoculation with A/Bat/Egypt/381OP/2017 (H9N2)

As shown by an indirect immunofluorescence assay, all directly inoculated animals sacrificed at day 21 had seroconverted against A/bat/Egypt/381OP/2017 (H9N2), whereas all contact animals remained seronegative (Table 3, Appendix A). Interestingly, only one out of two inoculated animals sacrificed at 7 dpi was seropositive against A/bat/Egypt/381OP/2017 (H9N2) (inoculated animal #5), while the two contact bats introduced at 1 dpi were seronegative (contact animals #5 and #6). These observations were substantiated by ELISA, where we were able to detect IAV-NP reactive antibodies in two out of four inoculated animals (inoculated animals #1 and #3 at 21 dpi) (Table 3, Appendix A). A third animal (inoculated animal #2) had a reaction conspicuously higher than the sera taken from naïve animals and was therefore categorized as “questionable”. A similar observation was also applicable to one IFA-seropositive animal (inoculated animal #4) euthanized at 21 dpi and to one out of two IFA-seropositive animals (inoculated animal #5) euthanized at 7 dpi. Here, both animals did not overcome the threshold to be placed from “negative” to “questionable” in the ELISA, although signals showed a higher reactivity than the non-inoculated contact animals (Appendix A). However, particularly concerning inoculated animal #4, this might also be explained by the observations in RT-qPCR, where we could only detect 58 viral genome copy numbers per mL, which does not necessarily lead to a strong seroconversion. Overall, these observations further document the successful infection of Egyptian fruit bats with A/bat/Egypt/381OP/2017 (H9N2).

## 4. Discussion

Our study shows for the first time that oronasal inoculation using LPAIV of subtype H9N2 (A/layer chicken/Bangladesh/VP02-plaque/2016) does not result in any virus replication or specific seroconversion in an experimental setup with Egyptian fruit bats. This suggests that these fruit bats are not susceptible to an avian-origin H9N2 influenza A virus infection. This was of special interest since there was, contrastingly, a clear indication of H9-specific antibodies in African fruit bats [12]. Therefore, the observed insensitivity against avian influenza of subtype H9N2 was unexpected but could have several reasons. On the one hand, the virus strain we used could be incompatible to Egyptian fruit bats. The Eurasian avian H9N2 lineage is divided into three different sub lineages, consisting of the Korean, also called Y439, lineage, the diverse Y280 lineage and the G1 lineage with the reference strain A/Quail/Hong Kong/G1/1997 [20,21]. The strain A/layer chicken/Bangladesh/VP02-plaque/2016 accounts to the latter lineage, which is extremely diverse and contains multiple sub lineages as well [22]. Thus, we cannot fully exclude that Egyptian fruit bats could be susceptible to a different avian H9N2 virus of the G1 lineage. However, it appears to be more likely that Egyptian fruit bats are simply not susceptible to avian H9N2 viruses coming directly from poultry without any adaptation processes. In contrast, in vitro replication studies using lung epithelial cells derived from three different bat species demonstrated a general permissiveness of these cells for other avian influenza viruses, e.g., subtype H6N1 and H2N3 [23]. On the other hand, the host species chosen to be inoculated might have been erroneous. Although specific antibodies against influenza A viruses of subtype H9N2 were demonstrated particularly in straw-colored fruit bats (*Eidolon helvum*) [12], this observation does not likewise guarantee replication of avian H9N2 in Egyptian fruit bats (*Rousettus aegyptiacus*). In addition, the infection dose of 10^4.7^ TCID_50_ per animal used in this experimental inoculation could be another factor, potentially in combination with the application route. However, oral swabs taken from naturally infected Egyptian fruit bats were more likely to be tested positive for viral RNA than rectal swabs, highlighting that oral epithelial cells of Egyptian fruit bats are able to replicate IAV [13].

### Bat Influenza Viruses May Be Highly Adapted to Particular Bat Species

Despite the fact that bat borne H9N2 showed strong genetic similarities to avian H9N2 viruses, the phylogeny revealed that in contrast to the HA and NA, all inner segments were very divergent from the recent AIV [13,14]. Therefore, this kind of bat flu virus might also exist in straw-colored fruit bats in Ghana and is the reason for the reported H9-seroreactivity [12].

Overall, although inoculation of Egyptian fruit bats with A/layer chicken/Bangladesh/VP02-plaque/2016 (H9N2) did not result in a productive infection of these animals, inoculation and thereby infection of the same species with A/bat/Egypt/381OP/2017 (H9N2) was successful. First indications for a successful infection were detected by a permanent temperature monitoring of four infected and four contact animals, which only showed subtle differences between both groups, that would have been unnoticed by monitoring of the rectal temperature alone. Therefore, monitoring of the body core temperature independent of any manipulation of the animal is important for challenge experiments where only mild or even no clinical signs are to be expected in the infected animals. Although we were not able to detect viral RNA related to A/bat/Egypt/381OP/2017 (H9N2) in oral/rectal swabs and organ samples from the direct contact animals, we demonstrated mucosal epithelium degeneration via histopathologic analysis in contact animal #6 (Figure 7), indicating that the virus was able to be transmitted from bat to bat. Furthermore, we could show that the isolation and propagation of A/bat/Egypt/381OP/2017 (H9N2) from three different Egyptian fruit bat samples of our experiment was successful in embryonated chicken eggs.

In addition to virus isolation and propagation, we observed that all animals directly inoculated with A/bat/Egypt/381OP/2017 (H9N2) and euthanized at 21 dpi displayed seroconversion. Although we could not demonstrate threshold-exceeding seropositivity of all animals in the ELISA that were seropositive in the IFT, we nevertheless saw obvious differences in the data values of these animals compared to the naïve animals (Appendix A). Taken together, these serological observations further support the evidence of a successful replication of A/bat/Egypt/381OP/2017 (H9N2) in Egyptian fruit bats after direct inoculation, and the RT-qPCR results moreover provide some evidence that the virus is able to be transmitted from bat to bat.

Recently published observations concerning the replication of another bat influenza A virus, H18N11, in various animal models proved that H18N11 is being transmitted among and is able to replicate in bats, whereas replication in mice and ferrets was only weakly detectable [24]. Thus, bat influenza A virus susceptibility, transmission and replication could be species-specific and bats in general might not be susceptible to influenza viruses originating from a non-bat host.

This hypothesis could thereby explain why the same species of flying foxes are susceptible to infection with A/bat/Egypt/381OP/2017 (H9N2), but not to A/layer chicken/Bangladesh/VP02-plaque/2016 (H9N2). However, due to the rapid adaptation processes of influenza viruses in general, the zoonotic potential of these viruses should not be underestimated and cannot be excluded without further experiments. Therefore, future studies will focus in more detail on the different influenza virus infections of fruit bats and the necessary adaptation processes. Our study provided first insights into the host range of different H9N2 viruses and the adaptation of bat-origin H9N2 to Egyptian fruit bats. Future studies will focus on the minimum infectious dose or the reactivity of the host following bat-origin H9N2 infection.

## Figures and Tables

**Figure 1 viruses-13-00672-f001:**
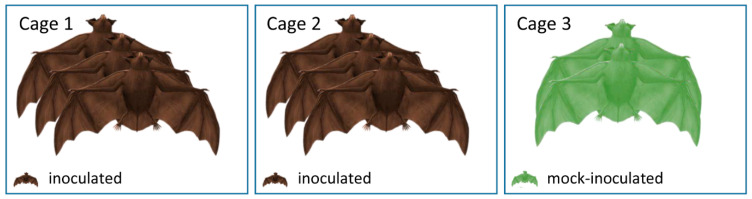
Experimental design of the experiment with “A/layer chicken/Bangladesh/VP02-plaque/2016 (H9N2)”. Three inoculated bats each were housed in two separate cages. The two mock-inoculated individuals were housed separately in cage 3.

**Figure 2 viruses-13-00672-f002:**
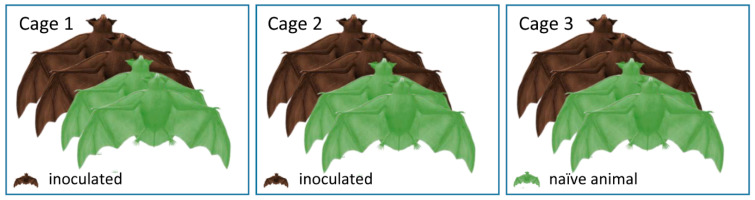
Experimental design of the experiment with A/bat/Egypt/381OP/2017 (H9N2). On 1 dpi, two inoculated and two naїve bats were mixed and housed in each of the three cages. This animal distribution was eventually maintained until the end of the experiment.

**Figure 3 viruses-13-00672-f003:**
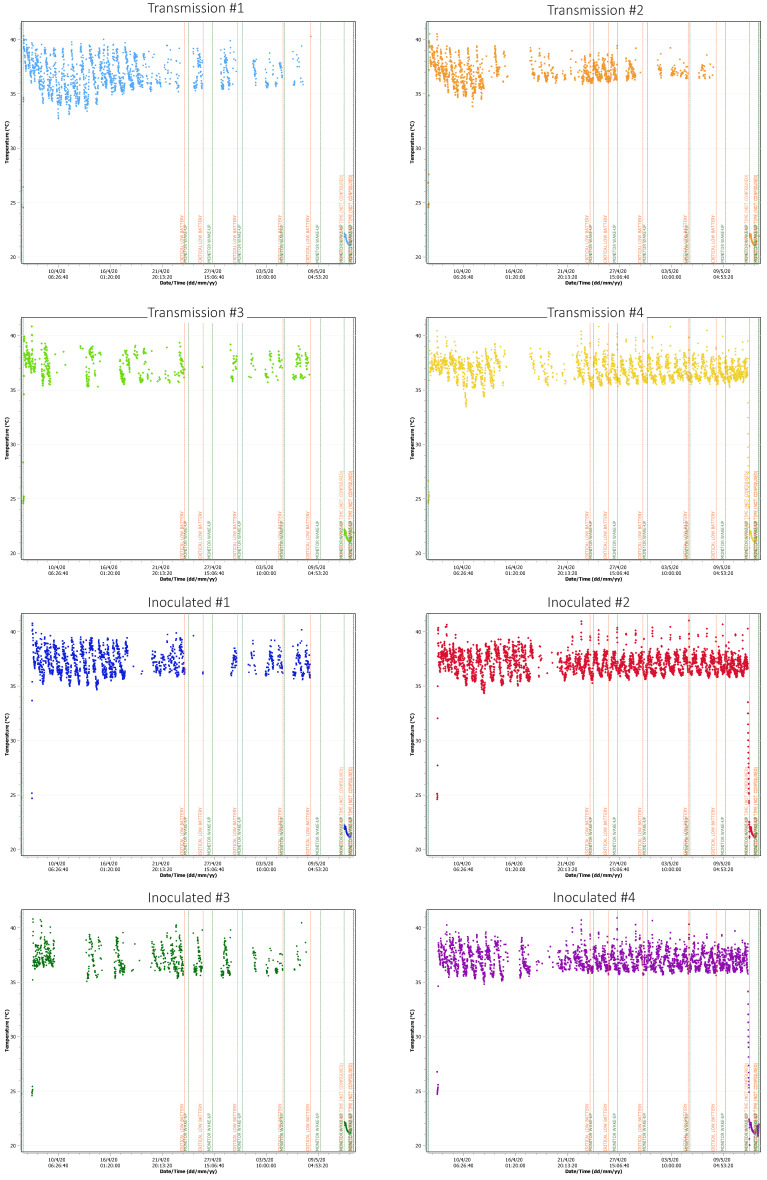
Temperature data of four infected and four contact animals. Body core temperatures were recorded every 15 min, some of the signals were lost due to the experimental setup (metal cages interfere with signal transduction to the monitor). Blue vertical line indicates challenge; red horizontal line shows 41 °C.

**Figure 4 viruses-13-00672-f004:**
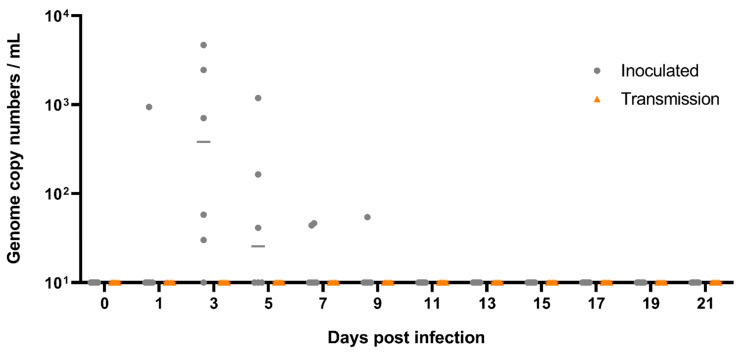
Viral shedding from oral swabs of the *Rousettus* bats following A/bat/Egypt/381OP/2017 (H9N2) infection. Viral shedding was monitored by RT-qPCR from swabs taken from the oral cavity. Swabs of six inoculated animals (grey dots) and six contact animals (orange triangles) were taken individually every other day. Samples below 10 viral genome copy numbers per mL (equivalent to Ct 39.2) were considered negative. DPI = Days post infection.

**Figure 5 viruses-13-00672-f005:**
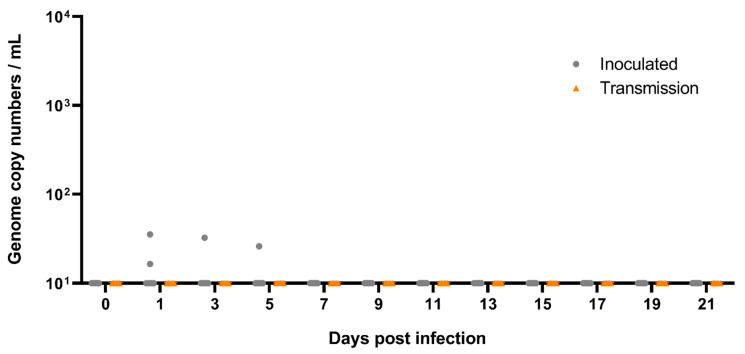
Viral shedding from rectal swab samples of the *Rousettus* bats following infection with A/bat/Egypt/381OP/2017 (H9N2). Viral shedding was monitored by RT-qPCR from rectal swabs. Swabs of six inoculated animals (grey dots) and six contact animals (orange triangles) were taken individually every other day. Samples below 10 viral genome copy numbers per mL (equivalent to Ct 39.2) were considered negative. DPI = Days post infection.

**Figure 6 viruses-13-00672-f006:**
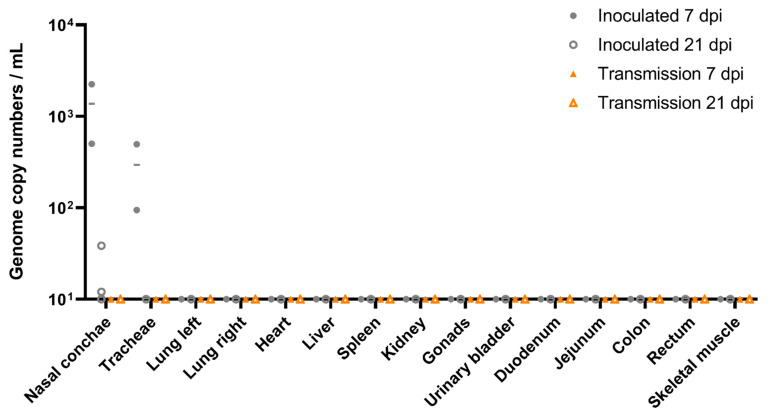
Viral loads in organs of A/bat/Egypt/381OP/2017 (H9N2)-inoculated Egyptian fruit bats. Individual results of detected viral RNA are depicted. Organs from two inoculated (filled grey dots) and two contact animals (filled orange triangles) were taken at 7 dpi, while organs from the residual four inoculated (open grey dots) and contact animals (open orange triangles) were taken after euthanasia at 21 dpi. Samples below 10 viral genome copy numbers per mL (equivalent to Ct 39.2) were considered negative.

**Figure 7 viruses-13-00672-f007:**
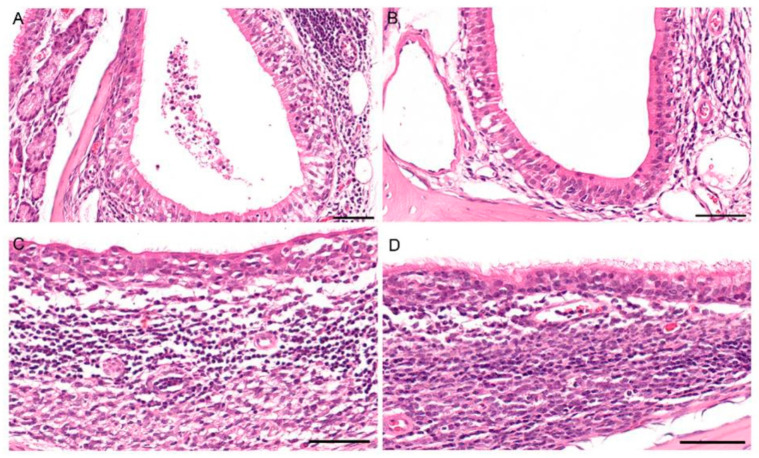
Histopathology in the nasal mucosa of fruit bats after infection with bat-origin H9N2 (A/bat/Egypt/381OP/2017), 7 dpi. (**A**) Inoculated animal #5: moderate transmigration of granulocytes and lymphocytes as well as intraluminal cellular debris within the nasolacrimal duct. (**B**) Unaffected nasolacrimal duct for comparison. (**C**) Contact animal #6: loss of cilia and moderate degeneration of the mucosal epithelium of the nasal-associated lymphoid tissue (NALT). (**D**) Unaffected NALT for comparison. All bars = 50 µm.

**Table 1 viruses-13-00672-t001:** Probes and primers for detection of A/bat/Egypt/381OP/2017 (H9N2) viral RNA (specifically designed for this study) and A/layer chicken/Bangladesh/VP02-plaque/2016 (H9N2) viral RNA.

Primer/Probe	Sequence	Concentration	Accession Number
Detection of A/bat/Egypt/381OP/2017 (H9N2) viral RNA			
H9N2-PB1-101Fv2	tga tcc acc cta cag cca tg	20 μM	MH376908 (Pos. 78-97)
H9N2-PB1-180Rv2	ctt ttt ctg aat att gat gag tcc ta	20 μM	MH376908 (Pos. 132-157)
H9N2-PB1-125FAMv2	FAM-tgg cac agg ata tac aat gga cac cgt-BHQ1	5 μM	MH376908 (Pos. 102-128)
Detection of A/layer chicken/Bangladesh/VP02-plaque/2016 (H9N2) viral RNA			
IAV-PB1_120F	cat ttg aat gga ygt caa ycc ga	20 μM	[17]
IAV-PB1_271R	ctg ttd acy gtg tcc atd gtg ta	20 μM	
IAV-PB1_247as_FAM	FAM-ccw gty ccy gty cca tgg ctg ta-BHQ1	5 μM	

**Table 2 viruses-13-00672-t002:** Summary of the isolation and propagation experiments of fruit bat samples inoculated with A/bat/Egypt/381OP/2017 (H9N2) in embryonated chicken eggs.

	Inoculated Animal #3 Oral Swab, 3 DPI	Inoculated Animal #6 Oral Swab, 3 DPI	Inoculated Animal #6 Nasal Conchae, 7 DPI
Initial Ct-value	30.11	28.90	31.74
Egg 1	31.68	28.64	34.61
Egg 2	31.82	14.58	13.55
Egg 3	34.11	14.05	13.47

**Table 3 viruses-13-00672-t003:** Summary of antibody detection from samples of six fruit bats inoculated with A/bat/Egypt/381OP/2017 (H9N2) and six contact Egyptian fruit bats.

Egyptian Fruit Bats ID	Immunofluorescence Assay */ELISA °		
	0 dpi	7 dpi	21 dpi
Contact animal #1	neg/neg	nd	neg/neg
Contact animal #2	neg/neg	nd	neg/neg
Contact animal #3	neg/neg	nd	neg/neg
Contact animal #4	neg/neg	nd	neg/neg
Contact animal #5	neg/neg	neg/neg	nd
Contact animal #6	neg/neg	neg/neg	nd
Inoculated animal #1	neg/neg	nd	pos/pos
Inoculated animal #2	neg/neg	nd	pos/quest
Inoculated animal #3	neg/neg	nd	pos/pos
Inoculated animal #4	neg/neg	nd	pos/neg
Inoculated animal #5	neg/neg	pos/neg	nd
Inoculated animal #6	neg/neg	neg/neg	nd

* Values considered positive at a dilution > 1:64; nd not done; ° values considered positive according to manufacturer’s instructions.

## Data Availability

The genome sequence generated in this study is available under the GISAID accession numbers EPI1807182-1807189. Data is contained within the article or Appendix A.

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
