# Peer review of "Egyptian Fruit Bats (Rousettus aegyptiacus) Were Resistant to Experimental Inoculation with Avian-Origin Influenza A Virus of Subtype H9N2, But Are Susceptible to Experimental Infection with Bat-Borne H9N2 Virus"

_viruses, 2021, doi:10.3390/v13040672_

Round 1

Reviewer 1 Report

The submitted manuscript concerns the pathogenicity of a zoonotic subtype of Influenza A virus H9N2 revealing the disprepancies in supsceptibility of Egyptian fruit bats to inoculation with avian origin and bat-born H9N2 viruses. The results provide not only a data on susceptibility of Rousettus aegyptiacus to H9N2 infection and also a new knowledge on risk transmission of H9N2 between different animals species.  The study is well designed and the manuscript is well prepared.

In the legend of Fig. 4 and 5  the negative samples are depicted as Ct 45 whereas the y axis shows Ct 40. Please correct viral load presentation.

Author Response

“The submitted manuscript concerns the pathogenicity of a zoonotic subtype of Influenza A virus H9N2 revealing the disprepancies in supsceptibility of Egyptian fruit bats to inoculation with avian origin and bat-born H9N2 viruses. The results provide not only a data on susceptibility of Rousettus aegyptiacus to H9N2 infection and also a new knowledge on risk transmission of H9N2 between different animal species. The study is well designed and the manuscript is well prepared.

We thank the reviewer for considering our study being well designed and well prepared.

In the legend of Fig. 4 and 5 the negative samples are depicted as Ct 45 whereas the y axis shows Ct 40. Please correct viral load presentation.”

Following the recommendation of Reviewer 2, we established a standard for our samples and changed our data presentation from Ct-values to viral genome copy numbers per ml. However, we are also thankful to the reviewer for pointing out to check for correct data presentation in each figure. We checked again to make sure every scale is in accordance with our figure legends in terms of data presentation.

Reviewer 2 Report

Halwe and colleague described the inoculation of Egyptian fruit bats (Rousettus aegyptiacus) with avian-origin Influenza A virus of subtype H9N2 and bat-borne H9N2 virus. They showed that Rousettus aegyptiacus are resistant to infection by avian H9N2 virus, no detection of the viral genome in  oral and rectal swab of the inoculated animals up to day 21 pi. Also, the animals did not serocoverted.  While the Rousettus aegyptiacus are susceptible to bat origin H9N2 strain, they authors showed that the animals were seroconverted and the viral genome was detected by qRT-PCR.

However I have  major concerns about the qRT_PCR presented in figures 4,5,6 and table 3

The authors presented data as ct values and showed the change in ct values overtime. This is inaccurate and could lead to false interpretation of the results especially if the ct >35

The authors need to show a quantitative measurement of the viral load over time by including a standard of know conc ( copy or IU) and correlate the viral load detected with the standard included in each run.

Also, what is the negative control used in each run?  

Author Response

“Halwe and colleague described the inoculation of Egyptian fruit bats (Rousettus aegyptiacus) with avian-origin Influenza A virus of subtype H9N2 and bat-borne H9N2 virus. They showed that Rousettus aegyptiacus are resistant to infection by avian H9N2 virus, no detection of the viral genome in oral and rectal swab of the inoculated animals up to day 21 pi. Also, the animals did not serocoverted. While the Rousettus aegyptiacus are susceptible to bat origin H9N2 strain, they authors showed that the animals were seroconverted and the viral genome was detected by qRT-PCR.

However I have major concerns about the qRT_PCR presented in figures 4,5,6 and table 3.

The authors presented data as ct values and showed the change in ct values overtime. This is inaccurate and could lead to false interpretation of the results especially if the ct >35.

The authors need to show a quantitative measurement of the viral load over time by including a standard of know conc ( copy or IU) and correlate the viral load detected with the standard included in each run.

Also, what is the negative control used in each run?

In this study, it was of particular interest to generally characterize this newly discovered virus in its natural host by analyzing susceptibility, pathogenesis and transmission semi-quantitatively. Therefore, we were initially not planning to quantitatively analyze our results, and therefore excluded the viral genome copy numbers from our manuscript.

Nevertheless, we agreed that it would be an improvement to show viral genome copy numbers based on a standard of known concentration, and we addressed this point now in the revised manuscript. A standard RNA (measured by digital droplet PCR) was established and the samples were re-run in RT-qPCR. We updated our figures in the manuscript accordingly. Negative controls were RT-qPCR reaction with no template. In addition, extraction controls (same matrix as samples from the animals) were included in the initial analyses.

Reviewer 3 Report

The article entitled, “Egyptian fruit bats (Rousettus aegyptiacus) were resistant to experimental inoculation with avian-origin Influenza A virus of subtype H9N2, but are susceptible to experimental infection with bat-borne H9N2 virus,” describes the susceptibility of Egyptian fruit bats to H9N2 viruses by inoculating bats with avian or bat H9N2 viruses as well as testing contact-transmission for bat H9N2. The results of the study suggests bat and not avian H9N2 viruses can sustain a productive infection in Egyptian fruit bats, but contact animals were not infected by bat H9N2. The manuscript has a modest impact on the current knowledge associated with IAV infection in fruit bats.

There are many points that require clarification/revisions:

In the Intro-add specific info about the avian H9N2 you chose to use.

Overall intro does not feel like it flows, and sentences are just mashed together. Please revise. For instance, the sentence in line 53/54 is talking about H9 seroreactivity but in the next sentence you talk about a new IAV virus (is it a H9 virus? Is it a complete new virus?).

Line 89-What antibodies were you testing for? Where other/similar HAs tested?

Line 91-What volume did you use to infect in mouth vs.nose? Equal volumes?

Line 93- Why did you use such a low quantity of virus? In the last study by your group, Gorka et al, you used 10^5.5 TCID50 in a similar infection.

Lines139-What was your PCR cutoff for positive versus negative?

Line 151- Collecting Flouresence values in the annealing step when you have an additional elongation step is highly usual. Please explain.  What were your positive/negative controls?

Table 2-Not necessary as long as you state nothing was detected or were all negative as you do in the preceding paragraph.

Lines 266-270-If your cutoff value is 37, it is not necessary to include the samples with values above that in your results because they are negative.

Line 279-282- You suggest many rectal swabs are positive but from Figure 5, it appears many were close to 40 which is beyond most conventional cutoff values.

Paragraph 308-322-Was there a problem with the inoculation for egg 1? Since both #6 oral swab and nasal conchae egg 1 values were much higher than the 2/3 replicates.

Table 4/Line 329-331-Please show pictures of positive versus negative IFA results and the S/P values for the ELISA results.

Line 360-361-Not sure this sentence has anything to do with the rest of the paragraph. Please rephrase or remove.

Line 361-362-Please explain why Rift Valley Fever Virus susceptibility has anything to do with IAV susceptibility?

Line 370-371-Please explain why strain variations are not the reason for the lack of seroconversion.

Line 373-374-Please remove the sentence about resistance. There is no evidence to support this.

Line 381-382- Since oronasal IAV inoculation is relatively common, please explain why it could be the reason for low replication or perhaps offer evidence that a different route might be more applicable here.

Line 388-391- You mention sequence differences but offer no specifics of what the differences are. Please include this information in your discussion.

Discussion- The discussion is too long and unfocused, plus there are a lot of false or exhaggerated statements. Please revise this entire section.

Author Response

The article entitled, “Egyptian fruit bats (Rousettus aegyptiacus) were resistant to experimental inoculation with avian-origin Influenza A virus of subtype H9N2, but are susceptible to experimental infection with bat-borne H9N2 virus,” describes the susceptibility of Egyptian fruit bats to H9N2 viruses by inoculating bats with avian or bat H9N2 viruses as well as testing contact-transmission for bat H9N2. The results of the study suggests bat and not avian H9N2 viruses can sustain a productive infection in Egyptian fruit bats, but contact animals were not infected by bat H9N2. The manuscript has a modest impact on the current knowledge associated with IAV infection in fruit bats.

There are many points that require clarification/revisions:

In the Intro-add specific info about the avian H9N2 you chose to use.

Following the reviewer’s suggestion, the specific clade belonging of the avian H9N2 was added. Further information is provided by naming the accession numbers of the corresponding sequences in the material and method section where interested readers will find more relevant details.

Overall intro does not feel like it flows, and sentences are just mashed together. Please revise. For instance, the sentence in line 53/54 is talking about H9 seroreactivity but in the next sentence you talk about a new IAV virus (is it a H9 virus? Is it a complete new virus?).

Since bat influenza viruses are just emerging and this topic is highly complex with multiple connections to other related influenza-viruses, there are numerous, but also not directly manuscript-related points that need to be addressed in the introduction, making it rather complex:

  1. The discovery of newly identified H18N11 and H17N10 as the first evidence of influenza A viruses originating from bats
  2. The demonstration of H9-seroreactivity in bats, indicating there might be potential H9N2 influenza A viruses circulating in bats
  3. The finding of bat-originating H9N2 IAV and its potential reassortment vessel.

However, we carefully revised the introduction following the reviewer’s suggestions (rearranging and rephrasing) and tried to make it easier to follow the relevant points.

Line 89-What antibodies were you testing for? Where other/similar HAs tested?

We tested for NP-reactive antibodies, in addition to H9 specific antibodies by VNT and H9N2 specific antibodies using the immunofluorescence assay (second experiment).

Line 91-What volume did you use to infect in mouth vs.nose? Equal volumes?

We inoculated the animals oro-nasally with 100 µl of respective viruses proportioned equally into each nostril and the mouth. We straightened this point out more clearly in the manuscript.

Line 93- Why did you use such a low quantity of virus? In the last study by your group, Gorka et al, you used 10^5.5 TCID50 in a similar infection.

In a former study, we used indeed a virus titer for inoculation that was six times higher than the one in this present study. This slightly lower titer was used in order to use the same dose of virus for the inoculation of animals with H9N2 of avian and H9N2 of bat origin.

Lines139-What was your PCR cutoff for positive versus negative?

After evaluation using a standard RNA (suggestion of reviewer 2) we have set the cutoff-value at 39.2; this would reflect 10 genome copies per mL We updated all our figure legends accordingly and stated that every sample undercutting 10 viral genome copies per mL were considered negative. Additionally, we have added a further sentence answering the question for clarification now also in the text itself, in addition to the figure legends.

Line 151- Collecting Fluorescence values in the annealing step when you have an additional elongation step is highly usual. Please explain. What were your positive/negative controls?

We used the CFX96 machine to analyze the samples. This cycler allows for collecting fluorescence values during the annealing and elongation step. In house testing revealed that curves detecting fluorescence in the annealing step look better.

In case of A/layer chicken/Bangladesh/VP02-plaque /2016 (H9N2), we performed the RT-qPCR, as it is mentioned in the text, according to Grund et al., 2018, targeting IAV-PB1 (17). This is a validated qPCR approach suitable for the detection of diverse Influenza viruses, including H18N11 and H17N10. The positive control in this case was RNA extracted from an A/swan/Germany/R65/2006 (H5N1) sample. The negative control was H2O.

However, when using these previously described primers and probe the assay was unsensitive for A/bat/Egypt/381OP/2017 (H9N2). Therefore, as mentioned in the manuscript, we had to specifically design primers and probe for A/bat/Egypt/381OP/2017 (H9N2). Thus, there was no positive control available before starting this experiment. However, we discussed this issue thoroughly, and now established a standard for our samples based on the original extracted viral RNA of our inoculum. We updated our figures and the manuscript text according to the newly calculated viral genome copy numbers per ml. The negative control again was H2O.

All RT-qPCRs were run also with the use of extraction controls.

Table 2-Not necessary as long as you state nothing was detected or were all negative as you do in the preceding paragraph.

We agree to the reviewer’s suggestion and deleted the table in the manuscript.

Lines 266-270-If your cutoff value is 37, it is not necessary to include the samples with values above that in your results because they are negative.

As mentioned before, after evaluation using a standard RNA (suggestion of reviewer 2) we have set the cutoff-value at 39.2, which reflects 10 viral genome copies per mL.

Line 279-282- You suggest many rectal swabs are positive but from Figure 5, it appears many were close to 40 which is beyond most conventional cutoff values.

As mentioned above, we have set the cutoff Ct-value at 39.2, which corresponds to 10 genome copies per mL.

Paragraph 308-322-Was there a problem with the inoculation for egg 1? Since both #6 oral swab and nasal conchae egg 1 values were much higher than the 2/3 replicates.

Inoculation of the eggs was performed in triplicates (3 eggs per sample) on the same day for all samples. We are not aware of any specialties concerning inoculation of egg #1. Rather, one possibility is that the detected Ct-values might represent the limit of viral load necessary to induce an infection. Another explanation would argue that a certain quasispecies was inoculated into egg #1 that does not replicate well in an embryonated egg. Therefore, we think this difference reflects effects coming from borderline evaluation and should not be overestimated.

Table 4/Line 329-331-Please show pictures of positive versus negative IFA results and the S/P values for the ELISA results.

As suggested, we generated a panel of photographs illustrating the IFA results and in addition showed the S/P values of the Elisa results. These have been added as Supplementary Figure 1.

Line 360-361-Not sure this sentence has anything to do with the rest of the paragraph. Please rephrase or remove.

During the suggested revision of the entire discussion section, this sentence was deleted.

Line 361-362-Please explain why Rift Valley Fever Virus susceptibility has anything to do with IAV susceptibility?

This sentence was deleted.

Line 370-371-Please explain why strain variations are not the reason for the lack of seroconversion.

This sentence was deleted during the revision of the discussion.

Line 373-374-Please remove the sentence about resistance. There is no evidence to support this.

This sentence was rephrased.

Line 381-382- Since oronasal IAV inoculation is relatively common, please explain why it could be the reason for low replication or perhaps offer evidence that a different route might be more applicable here.

We think oronasal inoculation was the right way to perform the experiment. However, to make to reader aware of variables that might have an influence we discuss different inoculation routes in the discussion section.

Line 388-391- You mention sequence differences but offer no specifics of what the differences are. Please include this information in your discussion.

The evaluation of sequence differences is beyond the scope of the discussion relevant to the topic of the manuscript. With the reference of Ciminski et al. 2020 the reader is referred to a relevant review article.

Discussion- The discussion is too long and unfocused, plus there are a lot of false or exaggerated statements. Please revise this entire section.

As suggested by the reviewer’s comments, we revised the discussion section.

Round 2

Reviewer 2 Report

I read carefully the replies of the authors to my comments and I do not have any further concerns.

Author Response

We happily appreciate that the reviewer sees all requests of the first review fulfilled.

Reviewer 3 Report

The revision of the article entitled, "Egyptian fruit bats (Rousettus aegyptiacus) were resistant to experimental inoculation with avian-origin Influenza A virus of subtype H9N2, but are susceptible to experimental infection with bat-borne H9N2 virus, greatly improved the quality and confidence in the findings of the manuscript. The article has a modest impact on the IAV field but found that directly infected but not contact bats can become infected with bat H9N2. 

There are a few minor points that require clarification:

Line 44: Rephrase "raised awareness"

Line 45: Which mammalian species? Be specific

Line 47: Change "is supposed to be" to "is"

Line 55: Rephrase "in correspondence to" as it doesn't make sense in the present context

Line 155: Change concentration to concentrations

Lines 155-156: Rephrase sentence as it is a bit confusing

Lines 216-217, 255-256, 262-263, 276-277: Rephrase "undercutting" as it does not make sense in the present form. Also, add the corresponding Ct to the 10 genomic copy cut-off. 

Lines 278, 282 Rephrase 'slightly positive'

Line 328 Change 'show seroconversion' to 'seroconvert'

Line 399 Change 'we could demonstrate' to 'we demonstrated'

Author Response

We thank the reviewer for taking the time again to thoroughly go through the manuscript.

We have adressed the final  requests als follows:

Line 44: Rephrase "raised awareness"

this was deleted

Line 45: Which mammalian species? Be specific

we added 'such as pigs or mice'.

Line 47: Change "is supposed to be" to "is"

this was changed as suggested

Line 55: Rephrase "in correspondence to" as it doesn't make sense in the present context

This was changed into 'furthermore'

Line 155: Change concentration to concentrations

changed as suggested

Lines 155-156: Rephrase sentence as it is a bit confusing

we changed this into: 'Quantification was established by...'

Lines 216-217, 255-256, 262-263, 276-277: Rephrase "undercutting" as it does not make sense in the present form.

we changed 'undercutting' for 'below'

Also, add the corresponding Ct to the 10 genomic copy cut-off. 

the corresponding Ct (39.2) was added

Lines 278, 282 Rephrase 'slightly positive'

we deleted 'slightly'

Line 328 Change 'show seroconversion' to 'seroconvert'

changed as suggested

Line 399 Change 'we could demonstrate' to 'we demonstrated'

changed as suggested